# Validation of a Body-Conducted Sound Sensor for Respiratory Sound Monitoring and a Comparison with Several Sensors

**DOI:** 10.3390/s20030942

**Published:** 2020-02-10

**Authors:** Takeshi Joyashiki, Chikamune Wada

**Affiliations:** 1Graduate School of Life Science and Systems Engineering, Kyushu Institute of Technology, 2–4 Hibikino, Wakamatsu-ku, Kitakyushu 808−0196, Japan; wada@brain.kyutech.ac.jp; 2Saiseikai Yahata General Hospital Department of Clinical Engineering, Harunomachi5-9-27, Yahatahigasi-ku, Kitakyusyu 805-0050, Japan

**Keywords:** body-conducted sound sensor, respiratory sound simulator, sensitivity evaluation system, bioacoustics

## Abstract

The ideal respiratory sound sensor exhibits high sensitivity, wide-band frequency characteristics, and excellent anti-noise properties. We investigated the body-conducted sound sensor (BCS) and verified its usefulness in respiratory sound monitoring through comparison with an air-coupled microphone (ACM) and acceleration sensor (B & K: 8001). We conducted four experiments for comparison: (1) estimation by equivalent circuit model of sensors and measurement by a sensitivity evaluation system; (2) measurement of tissue-borne sensitivity-to-air-noise sensitivity ratio (SRTA); (3) respiratory sound measurement through a simulator; and (4) actual respiratory sound measurement using human subjects. For (1), the simulation and measured values of all the sensors showed good agreement; BCS demonstrated sensitivity ~10 dB higher than ACM and higher sensitivity in the high-frequency segments compared with 8001. In (2), BCS showed high SRTA in the 600–1000 and 1200–2000-Hz frequency segments. In (3), BCS detected wheezes in the high-frequency segments of the respiratory sound. Finally, in (4), the sensors showed similar characteristics and features in the high-frequency segments as the simulators, where typical breathing sound detection was possible. BCS displayed a higher sensitivity and anti-noise property in high-frequency segments compared with the other sensors and is a useful respiratory sound sensor.

## 1. Introduction

Auscultation of respiratory sounds has long been an important physical assessment technique for medical workers [1], and with the influx of recent technology, auscultated respiratory sounds for pneumonia [2], bronchial asthma [3], and sleep apnea syndrome [4] are being used for chronic obstructive pulmonary disease (CODP) [5,6] diagnosis, tidal volume estimation [7], snoring detection [8], and apnea detection [9]. Moreover, respiratory rate monitoring using respiratory sounds [10] and snoring sensors [11] has become plausible due to unique advantages such as being a non-invasive technique and being based on low-cost, high-performance microphones that readily available in the market. Although the above-mentioned methods using respiratory sounds have been extensively studied via signal processing [3,4,5,6,7,8,9], the outcomes of these analyses have not often been practically implemented.

Respiratory sound sensor performance is a very important criterion [12,13,14,15] in signal processing. Some of the most commonly used sensors in post-research include air-coupled sensors (ACM: air-coupled microphone) [4,5,8,16] and acceleration sensors [7,17]. ACM uses a microphone to detect pressure in the air chamber, whereas an acceleration sensor directly detects skin vibrations via physical coupling. In the case of ACM, the influence of extraneous noise and the decrease in sensitivity due to the collapse of the air chamber’s seal must be taken into account. Such factors can even impede the use of ACM in long-term monitoring. Acceleration sensors are relatively expensive and suffer from contact-specific friction noise. Therefore, it is difficult to perform stable detection of bioacoustic signals via ACM and acceleration sensors reported in previous works. In addition, even though these sensors have flat frequency characteristics and high sensitivity over a wide range under normal usage, these characteristics undergo a significant changes when used on soft living tissue [15]. Thus, respiratory sound research may advance with the associated development of sensors suitable for detecting bioacoustic signals.

An important factor required for respiratory sound sensors is the ability to record weak respiratory signals and high signal-to-noise ratios. This paper focuses on body-conducted sound sensors (BCSs, Figure 1) [18]. BCS uses an acoustic propagation layer material that is acoustically matched to soft living tissue. Originally, BCS was developed for recording a non-audible murmur (NAM) [19]. Its application was subsequently extended to verbal communication [20,21] while also preserving its sensitivity to capture pulse and respiratory sounds other than NAM [22]. However, there is no comparative study on the utilization of BCS versus conventional sensors, and the usefulness of BCS remains unverified, especially in terms of measured physical quantities for both types of sensors. In particular, as BCS is a contact-type sensor with a built-in microphone, it is necessary to evaluate its sensitivity under the same physical coupling conditions as an acceleration sensor [23].

To address this gap in understanding, we compared BCS, ACM, and acceleration sensors in the hope of (1) estimating and comparing the frequency characteristics of each sensor, (2) comparing their sensitivity for soft-tissue and air-mediated signals, (3) comparing the sensors via a sound simulator, and (4) comparing actual recorded respiratory sounds of each sensor. The measurements were made under similar conditions with a sensitivity evaluation system and a respiratory sound simulator. The most important problem is the change in the frequency response of the sensor sensitivity when used on soft living tissues. This affects diagnosis and flow estimation. We used a sensitivity measurement evaluation system [23], lung sound simulator, and actual respiratory sounds to verify the characteristics of the comparisons against the theoretical values, and furthermore, to examine the usefulness of BCS for respiratory sound measurements.

## 2. Materials and Methods

### 2.1. Sensors of Interst

For comparison purposes, we used an ACM and an acceleration sensor (B & K: type 8001 [24]) alongside the BCS. Photographs and structural schematics of the three sensors are shown in Figure 2, and their specifications are presented in Table 1. The BCS was equipped with a remodeling microphone that had an exposed diaphragm (Primo: EM258N-remodelling); its housing was made of acrylonitrile-butadiene-styrene (ABS) resin, and a urethane elastomer (Exseal: Hitohada-gel) was used for the area between the microphone and the cylindrical soft living tissue (acoustic propagation layer) which had a diameter of 20 mm and a height of 1 mm. The ACM had a built-in, all-purpose microphone (Primo: EM258N [25]) and a housing that was also made of the ABS resin. A coupler was formed into the 20-mm-diameter, 1-mm-height cylindrical air chamber which had a skin contact surface width that was designed to be 3 mm in order to keep the air chamber closed. BCS and ACM had identical air chamber volumes. In the case of the acceleration sensor, a type 8001 sensor which was connected to a dedicated amplifier ((B & K: type 2635) was used. Type 8001 is an impedance head-tapped 10-32 UNF platform containing an accelerometer mounted on top of a force gauge. This allows for the simultaneous measurement of force and acceleration parameters. The impedance heads offer a simple approach to the measurement of point mechanical mobilities and impedances. They can be used on a wide range of structures, including rotor blades, polymers, rubbers, the human body, artificial mastoids, fruit, welds, wood, and metal panels [24].

### 2.2. Estimation of Frequency Characteristics and Measurements for Sensors on Soft Tissue

#### 2.2.1. Estimation Model for the Frequency Characteristics of Sensors

Sensors placed on soft living tissues exhibited a variation in their frequency characteristics. We tried to estimate the frequency characteristics of BCS using an equivalent circuit model; we also investigated the influential factors in a previous study [23]. Similarly, an investigation was carried out to evaluate the best frequency estimation methods for use on ACM and acceleration sensors [15]. The theoretical equations employed in this study to estimate the frequency response of investigated sensors are listed in Table 2. For equation modelling of the acceleration response, $Z_{r}$ is the mechanical impedance of soft living tissue [26], whereas the parameters are the radius of the contact surface of the sensor, the shear modulus, the shear viscosity, and the density of the chest wall.$\;Z_{M}$ is the mass of the sensor expressed by $Z_{M}=j\omega M$. In the case of the pressure response model, $Z_{m}$ indicates the impedance of the cylindrical air chamber compliance in ACM. Here, placing the ACM on the unloaded chest-wall surface affects the mass load of the sensor. Therefore, $Z_{M}$ is also taken into consideration. In addition, $A_{0}\left(\omega \right)$ represents the acceleration frequency characteristic of soft tissue at 1 Pa. Here, the ACM theoretical equation is the surface-free state case that assumes a state in which the sensor is placed continuously for a long time in soft living tissue, which is different from the fixed-surface state case in previous studies [15]. The implications of changes of the surface in a fixed-state case scenario will be examined in the Discussion section. For the BCS and acceleration sensors, the acceleration obtained by the sensors can be converted to pressure, which can be determined by multiplying the sensor mass/contact area (m/S) as in Table 2. These theoretical equations can be used to demonstrate the accuracy of the measurements.

#### 2.2.2. Measurements for the Sensors

We measured the acceleration sensitivity and pressure sensitivity of the sensors through the aid of a sensitivity measurement evaluation system [23] in the measurement frequency range of 100–2000 Hz [27], as well as in the required range for respiratory sound measurement. We used a small acceleration sensor (Primo: S15S5C) as the reference sensor in the measurement system to determine the acceleration voltage sensitivity (V/m/s^2^). Note that the accelerometer had a flat acceleration response at frequencies up to 2000 Hz. Subsequently, we converted the acceleration voltage sensitivity into pressure voltage sensitivity (V/Pa) through the product of the sound pressure frequency characteristics in soft living tissue [23]. We evaluated the sensors based on two experiments: (a) a comparison of the pressure voltage sensitivity (V/Pa) in BCS and ACM sensors and (b) a comparison of the acceleration voltage sensitivity (V/m/s^2^) in BCS and type 8001 sensor.

### 2.3. Tissue-Borne Sensitivity-to-Air Noise Ratio (SRTA)

An ideal sensor displays high tissue-borne sensitivity and low (or null) response to air-borne noise-transmitted sounds [14]. We calculated the ratio of tissue-borne sensitivity to the air-noise sensitivity ratio (SRTA) using noise loading experiments. We calculated the measurement sensitivity (V/Pa) for BCS and ACM based on the results in Section 2.2.2 and converted the type 8001 acceleration voltage sensitivity (V/m/s^2^) to pressure voltage sensitivity (V/Pa) via the product of the sound pressure frequency characteristics in soft living tissue. In the air-borne noise characteristic experiment (Figure 3), we used the measurement system employed in the sensitivity evaluation experiment and mounted the sensors on a urethane elastomer to simulate the soft tissue of a living body. Herein, white noise was generated by a speaker (audio-technical: AT-MSP5TV) from a distance of 0.3 m [14]. Such white noise was measured by the reference microphone and corrected to the air noise sensitivity every 1 Pa.

### 2.4. Evaluation of Sensors in the Respiratory Sound Simulator

To evaluate the actual respiratory sound, we created a respiratory sound simulator with reference to the bioacoustic transducer test system [28] that was created in a previous study. The respiratory sound simulator is shown in Figure 4. It had a headphone loudspeaker (JVC: HA-S500) that was sealed to the underside of the air chamber, and the space behind the speaker was filled with acoustic insulating material to attenuate any chamber resonances. Moreover, we created a cylindrical air chamber for the system in which a reference microphone (Primo: EM-258N) was inserted into the upper central position. Additionally, we drilled several holes, approximately 2 mm in diameter, in the upper part of this air chamber to transmit the sound from the headphones. We placed a urethane elastomer on the front of the air chamber as a propagation layer and a platform for the objects to be measured.

The sensors were connected to an audio interface (Roland: Rubix22). The speaker and reference microphone were calibrated using a 300-Hz sine wave. The frequency characteristics of the reference and sensors were measured in the range of 100–2000 Hz. The audio interface signal was input into a PC and was analyzed using the respiratory sound analysis software EasyLSA [3,4]. The respiratory sound sources that were measured included (1) white noise, (2) tracheal sounds [7,8,9], and (3) polyphonic wheezes [16] which were used for educational purposes included on the website [29]. Moreover, these sources were evaluated using spectrographic analysis and frequency characteristic analysis. Tracheal sound and polyphonic wheezes were divided into inspiratory and expiratory. Also, each frequency segment (100–200, 200–400, 400–800, 800–1600 Hz) referred to a previous study [5,30,31]. Here, the median and interquartile ranges (25–75%) were calculated. The reason for this frequency segmentation is that the respiratory sound has a main component of 400 Hz or lower, whereas the main component of a muscle sound may be 200 Hz or lower [32]. It is said that a frequency of 400 Hz or higher correlates well with the flow of breathing. Furthermore, wheezing sounds appear at 800 Hz or higher.

### 2.5. Actual Respiratory Sound Measurements

Finally, the sensors were evaluated through the actual respiratory sound measurements for 15 healthy young subjects. The test subject parameters are shown in Table 3. These experiments were approved by the Ethics Committee of Kyushu Institute of Technology. The actual measurement examination was performed in a quiet laboratory environment wherein the temperature and humidity were maintained constant. The measurement position was adopted from previous studies [4,7] in which the BCS was placed on one side of the subject's neck whereas the ACM or type 8001 sensor was placed on another side, as indicated in Figure 5a. Signals from the sensors were connected to an amplifier, analogue-to-digital (AD), and they were then converted using an audio interface and input to a PC. Analysis was performed using Easy LSA (Figure 5b). We conducted three measurements for comparison: (1) normal breathing tracheal sounds; (2) normal breathing tracheal sounds in a noisy environment; and (3) wheezing sounds from oral transmissions. Measurements for (1) were conducted for the breathing tracheal sound after breath training exercises. Measurements for (2) were conducted for the tracheal sound with a noise load under the same conditions as those defined in Section 2.3. The white noise used was 50 dB with a sound pressure level that was similar to that of daily conversation sound pressure levels. Measurements for (3) were conducted using the mask in Figure 5c with built-in speakers (JVC; HA-S500). Abnormal respiratory sounds were output from the speaker via the oral cavity side attached to the test subject [33,34] and were detected by the target sensor (Figure 5d). The sound source used was the polyphonic wheezes described in Section 2.4.

Data analysis for each experiment was conducted as follows: (1) the detected respiratory sound signal was divided into inspiration and expiration, and the box plot consisted of the 100–200, 200–400, 400–800, and 800–1600 Hz frequency intervals. Statistical analysis was performed using the Wilcoxon test. In (2), the power spectrum of the noise and inspiration/expiration of the respiratory sounds detected in (1) were compared for each sensor. In (3), the peak for the wheezing sound observed in each frequency segment (400–500, 500–600, and 1000–1200 Hz) were compared in the box plot. Statistical analysis was performed using the Wilcoxon test or Mann–Whitney U test. From all the data collected for the test subjects (n = 49 for BCS and ACM; n = 51 for BCS and type 8001), the presence or absence of the wheezing noise, as determined via a visual spectrogram, was defined as the sensitivity. This sensitivity was determined by calculating as follows: true positive (TP)/true positive + false negative (FN). Here, cases in which there was a recognizable wheezing signal were assigned as TP, and cases where there was no recognizable wheezing signal (i.e., it was unclear in the spectrogram) were listed as FN.

## 3. Results

### 3.1. Simulation and Measurement of Frequency Characteristics of Sensors

The simulated and measured frequency characteristics are shown in Figure 6. Here, the vertical axis denotes the sensitivity (voltage/pressure sensitivity [V/Pa]) (Figure 6a) or voltage/acceleration sensitivity [V/m/s^2^]) (Figure 6b), and the horizontal axis is the frequency. The dotted line shows the simulation result, and the solid line shows the measured value. In both simulation and measurement values, the errors had a consistent tendency to appear in the low-frequency segments.

The frequency response of the voltage/pressure sensitivity of BCS and ACM is shown in Figure 6a. BCS exhibited higher sensitivity than ACM in all frequency segments and was especially high in the 100–300-Hz segment. Further, frequency characteristics of BCS and the voltage/acceleration sensitivity of type 8001 are presented in Figure 6b. Type 8001 demonstrated a higher sensitivity, even though such sensitivity was attenuated with an increase in frequency. In contrast, BCS displayed peaks at nearly 200 and 400 Hz. The other sensors also exhibited stable sensitivity parameters in this 400-Hz segment.

### 3.2. SRTA Results

Results for the SRTA are illustrated in Figure 7. Here, the vertical axis is the SRTA, and the horizontal axis is the corresponding frequency. Apparently, BCS had higher SRTA than other sensors in the 600–1000 and 1300–2000-Hz segments. No difference in the values was observed for BCS and type 8001 sensor in the 100–600- and 1100–1200-Hz segments.

### 3.3. Observation Experiment using a Respiratory Sound Simulator

Measurement results using the respiratory sound simulator are presented in Figure 8, with the spectrogram (a) and frequency characteristics (b) for white noise, tracheal sounds, and polyphonic wheezes shown from left to right. For the spectrogram, the vertical axis indicates frequency, while the horizontal axis shows time. For the frequency characteristics, the vertical axis indicates the sound pressure level (SPL (dB)), while the horizontal axis shows frequency. For the tracheal sounds and polyphonic wheezes, the black line indicates the inspiratory data, whereas the orange line describes the expiratory responses.

Referring to the spectrogram, the maximum power value of the white noise from the 100–2000-Hz segment was approximately 92 dB (Easy LSA software) in BCS. On the other hand, in the case of ACM, it was 86 dB. In the 200–400-Hz segment, BCS was higher than ACM by approximately 9 dB. Type 8001’s maximum power value was 93 dB. There was maximum value in the 400 Hz segment; however, subsequent to this, the power value sharply dropped to 75 dB. Moreover, for tracheal sounds in the 200–400-Hz segment, the BCS detection was about 11–12 dB higher than that of ACM. For the type 8001 sensor, the signal level was about 82.5–87.5 dB in the low-frequency segment, i.e., 100–200 Hz; however, as the power value decreased as the frequency increased, the sensor’s signal level was approximately 72.5–74.0 dB in the 400–800-Hz frequency range. In cases where wheezing sounds were measured, for the signal of exhalation (Figure 8a ① and ②), BCS was able to clearly detect the source (63.5–74.0 dB). Likewise, the signal of the exhalation in ACM was noticeable, although it was weak at approximately 52.0–60.0 dB, whereas the type 8001 sensor had detection difficulties due to interference from other signals.

Based on the observations of the frequency characteristics of the sensors, BCS showed the characteristics for white noise in the original sound, whereas the low band in the ACM sensor was more emphasized. The type 8001 sensor had noticeable responses in the low-frequency segments, which then sharply dropped. Accordingly, the tracheal sound spectrogram of the respiratory sound signal was strongly detected in the low-frequency segments for the type 8001 sensor but was weakly detected in the high-frequency segments. Moreover, BCS detected higher frequency signals beyond 1000 Hz. A spectrogram of the wheezing sounds confirmed the presence of the characteristic signals observed at 1000 Hz or higher in the BCS sensor (Figure 8b ③). ACM and type 8001 also demonstrated such detection capabilities, albeit at a weaker signal level than BCS.

The power median (25–75%) values (dB) of each frequency segment in the inspiration and expiration portions of tracheal sounds and wheezes is shown in Table 4. In particular, the frequency segments were 100–200, 200–400, 400–800, and 800–1600 Hz, of which the latter two yielded the highest power median values for the BCS sensor.

### 3.4. Actual Respiratory Sound Observation

A box plot comparing the tracheal breathing sounds of 15 test subjects is shown in Figure 9. Here, Figure 9a,b are comparisons between the BCS and the ACM sensors, whereas Figure 9c,d are comparisons between the BCS and type 8001 sensor. Inspiration is shown on the left (a,c), whereas expiration is on the right (b,d). The vertical axis is defined as the power (dB), and the horizontal axis contains the frequency segments which were 100–200, 200–400, 400–800, and 800–1600 Hz. A comparison of BCS and ACM showed that BCS had a higher power value within all frequency ranges. A comparison of BCS and type 8001 showed that although type 8001 was more sensitive, it significantly attenuated with an increase in frequency. The inspiration cycle in the 800–1600-Hz segment and the expiration cycle in the 400–800-Hz section showed no significant differences. 

Figure 10 shows the results of noise application. Here, the vertical axis represents power (dB), and the horizontal axis is frequency. The dotted line indicates noise, the solid line indicates inspiration, and the double line indicates expiration. At a BCS value of less than 1000 Hz, the respiratory sound signal was higher than the noise level. In the case of ACM, the respiratory sound signal was stronger than noise signals that were less than 500 Hz; however, the noise signal was generally higher than the respiratory sound signal. In the case of the type 8001 sensor, the respiratory sound signal component was strong; however, the noise signal was high, especially when the frequency was less than 500 Hz. 

Figure 11 shows the peak intensity of the wheeze signal and the detection sensitivity. Here, BCS and ACM had a stronger power value in the target section in comparison to the values associated with ACM. Furthermore, the detection sensitivity of BCS was higher than that of ACM. A comparison between BCS and type 8001 showed that there was no significant difference between the results obtained at 400–500 Hz and 500–600 Hz. However, at 1000–1200 Hz, the type 8001 sensor could detect only one wheeze (1/51). The sensitivity of type 8001 was 76.47% (under 1000 Hz) and 1.96% (over 1000 Hz), which was lower than that of ACM (under 1000 Hz: 93.88% and over 1000 Hz: 34.69%). 

A representative case for the respiratory sounds that were actually observed are described in Figure 12. Here, (1) shows normal breathing tracheal sounds [normal tracheal sound], (2) shows normal breathing tracheal sounds in a noisy environment [+ white noise], and (3) shows wheezing sounds from mouth transmission [wheezing]. Between BCS and ACM sensors seen in Figure 12a, the former detected signals under normal tracheal sound observations. On the other hand, the white noise environment signal was weak, and the noise signal was strong in ACM (Figure 12a ①). The wheeze signal was observed for the high segments in the BCS sensor (Figure 12a ②); however, it was weakly detected in ACM.

Between the BCS and type 8001 sensor seen in Figure 12b, the former demonstrated detection capabilities in the high-frequency segments (Figure 12b ③). In contrast, type 8001 exhibited the ability to mix white noise in the low-frequency segments and had an unclear boundary between the inspiratory and expiratory periods (Figure 12b ④). For the wheezing noise, BCS could clearly detect the wheeze signal in the high-frequency segments (Figure 12b ⑤).

## 4. Discussion

### 4.1. Frequency Characteristic Results of Sensors

With respect to the BCS and ACM frequency characteristics, the BCS sensitivity was higher than that of ACM, indicating the effect exerted by the acoustically propagating layer and the reordering of the microphone. Thus, if the conditions in this experiment, i.e., a cylindrical housing shape with a diameter of 20 mm and height of 1 mm, the ABS resin housing, the housing shape, and the materials, were the same, then the BCS sensor could exhibit high sensitivity under these experimental conditions. Conversely, the ACM frequency characteristics were dependent on the conditions [15] described in 2.2.1.

In the case of the skin surface vibration measurements, two cases were considered, i.e., the surface-free state and the fixed-surface state. In the surface-free state, the surface vibration is measured without any load, whereas a sufficient load is applied to the ACM in the fixed-surface state, and the contact surface is completely restrained. In a previous study [15], the frequency characteristics of the ACM sensors when tested on soft living tissue in the fixed-surface state were investigated. This was similar to the scenario in which a medical doctor temporarily places a stethoscope on a patient’s chest wall. Given this, the objective of this study was to examine the measurement conditions by assuming continuous monitoring of the respiratory sounds [4,16]. In this study, we compared the frequency characteristics of the BCS and ACM sensors based on the respective pressure responses in the surface-free state obtained in soft living tissue. Therefore, it can be stated that BCS was better than ACM with respect to frequency responses under long-term conditions. 

For the frequency characteristics of the BCS and type 8001 sensor, the latter showed higher sensitivity below approximately 300 Hz. In a previous study, the main component of respiratory sounds was observed above 300 Hz [35]. In particular, skeletal muscle-derived components (muscle sounds) were included in the frequency segments under 200 Hz [32]. Therefore, if the sensitivity in the low-frequency region was high, then sounds other than the respiratory sounds were mixed in. As such, the type 8001 sensor’s high sensitivity only in the low region was considered unsuitable for respiratory sound monitoring. By contrast, although BCS showed a peak at 400 Hz, it had relatively flat characteristics even in the higher frequency segments, which made detection of respiratory sound components with high sensitivity possible.

### 4.2. Evaluation of SRTA Results

BCS demonstrated high SRTA in relatively high-frequency segments; this could be attributed to the effect of the urethane elastomer used in the acoustic propagation layer. On the contrary, ACM had a low value which was attributed to the effects of noise mixing as the sounds entered the air chamber. The acceleration sensor type 8001 without an air chamber showed a similar value as the BCS sensor at low frequencies. BCS demonstrated the highest sensitivity of all the sensors beyond 600 Hz, indicating its advantage over the others. This was also advantageous for the characteristic respiratory sounds, especially for wheezes that were detected at relatively high frequencies.

### 4.3. Simulator-Observed Respiratory Sounds

Observations performed using the respiratory sound simulator were similar to the results of the sensitivity measurement evaluation. The acceleration sensor type 8001 displayed the highest sensitivity for white noise detection in the low-frequency segments between 100 and 400 Hz. However, BCS displayed the highest sensitivity in the frequency segments higher than 400 Hz. Such a feature may be useful in the detection of characteristics of respiratory sounds within the 1000 Hz segment, for example, for applications involving the diagnosis of asthma or chronic obstructive pulmonary diseases (COPD) [5,36,37] as these are characterized by polyphonic wheezes.

The simulator was originally used for medical training [38]. Therefore, the evaluation of respiratory sound using this device yielded more accurate results. However, the standards for simulators and sensitivity evaluations have not been established, and the reference parameters for sensors and measurement conditions have not been unified [14,15,18,28,39]. A unified measurement standard is thus required. The simulator and sensitivity evaluation system used in this study increase the possibility for future improvements. 

### 4.4. Observation of Actual Respiratory Sound and Usefulness in Clinical Applications

The experimental procedure in which the sensors were placed on the subject's neck yielded results that were consistent with the frequency characteristics and respiratory sounds observed using the simulator. In BCS and type 8001 noise experiments, the former detected noise in the region above 1000 Hz owing to the sensor’s high tissue-borne sensitivity. Moreover, noise may be transmitted either directly to the sensor or through tissue. In the case where noise was directly transmitted to the sensor, BCS exhibited higher anti-noise properties than ACM. In the case where noise was transmitted though the tissue, BCS showed higher sensitivity. Among the sensors, BCS had the best detection capabilities for wheezing sounds within the high-frequency segments.

The detection of apnea in monitoring the respiration rate [4,9,10] and the detection of low sounds such as snoring [8] did not necessary require detection within the high-frequency segments. Typical sensors used in clinical practice included a snore sensor (Nihon Koden: TM-106A) attached to polysomnography [11], a film piezoelectric element [40], and a respiration rate-monitoring sensor (Masimo: Rad-87) [10] in which a vibration plate was attached to the inside of the sensor to detect skin surface vibrations. Both sensors are contact types and are currently being used for low-frequency segment signal detection.

Observations of the wheeze experiment performed in this study were done with the aim of obtaining characteristic signals in high-frequency segments; therefore, a wide-segment and high-sensitivity sensor was best suited for this task. Moreover, although still in the research stage, BCS is considered to be most useful for flow estimation of tracheal sounds [7] and for the diagnosis of respiratory diseases [4,8,9,16]. In these areas, spike noise [18,39] and non-signals that are partly integrated as noise are particularly important issues to overcome.

Furthermore, the characteristics of tracheal sounds which must be investigated by placing the sensors on other body parts and the frequency segments to be measured should be verified. For example, COPD patients are said to have increased high-frequency components in their respiratory sound [5]. BCS’s wide frequency range may prove to be useful in frequency regions that are higher than those previously considered.

In the future, it will be necessary to conduct long-term observation experiments using BCS. The timeframes for these observation experiments could be, for example, 6–8 h per night for the diagnosis of sleep apnea syndrome and a few days for the purpose of bedside monitoring in a hospital. As for now, we are unaware of the changes that may occur in the sensitivity of the instrument subsequent to prolonged use. Furthermore, it is necessary to investigate the sensitivity of the sensor’s detector in sub-optimal placement situations. Although this study focused only on the sensitivity of the sensor, sensor characteristics have many parameters (e.g., directivity and drift). Therefore, a comprehensive evaluation of these parameters would be required, and it is necessary to unify the evaluation systems for these bioacoustic sensors to set evaluation standards in the future.

## 5. Conclusions

The results of this study clarified certain areas as follows:(1)Given the same acoustic propagation layer shape, BCS had sensitivity higher than ACM. BCS showed high sensitivity in the high-frequency segments when compared with the acceleration sensors similar to type 8001.(2)BCS showed high SRTA in the high-frequency segments of 600–1000 and 1200–2000 Hz compared with the other sensors.(3)The simulator results demonstrated characteristics close to those of the acceleration response. BCS was advantageous for characteristic signal detection in the high-frequency segments.(4)In the tracheal sound observation experiment, the spectrogram for each sensor showed the same results as those of the simulator, which indicated the advantages of characteristic respiratory sound detection at high frequencies.

The case where BCS was placed on an actual living body is a future direction for this study. Here, we could use the respiratory sound, but we will consider applying other bioacoustics. The software Easy LSA used in the analysis is also operable on smartphones; thus, we expect it to be integrated into BCS sensors, not only for clinical applications, but also for mobile monitoring of health.

## Figures and Tables

**Figure 1 sensors-20-00942-f001:**
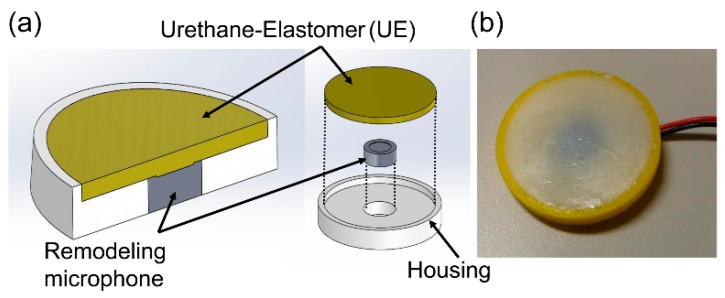
(**a**) Structure and (**b**) actual photograph of a BCS.

**Figure 2 sensors-20-00942-f002:**
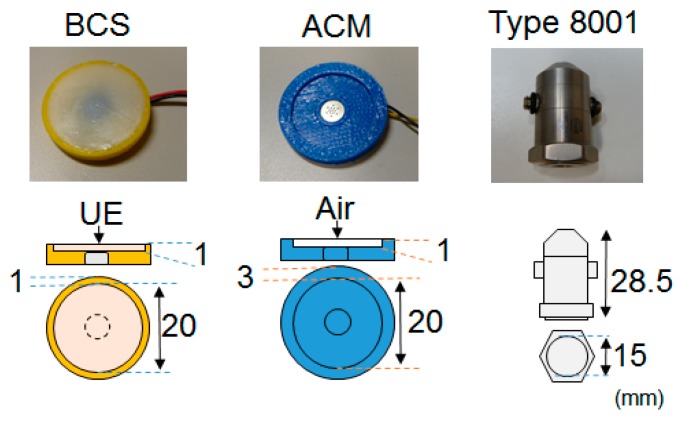
Photographs and structures of the sensors.

**Figure 3 sensors-20-00942-f003:**
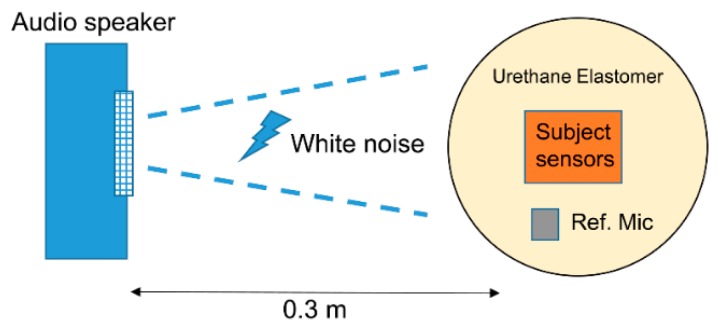
Air-borne noise characteristic experiment.

**Figure 4 sensors-20-00942-f004:**
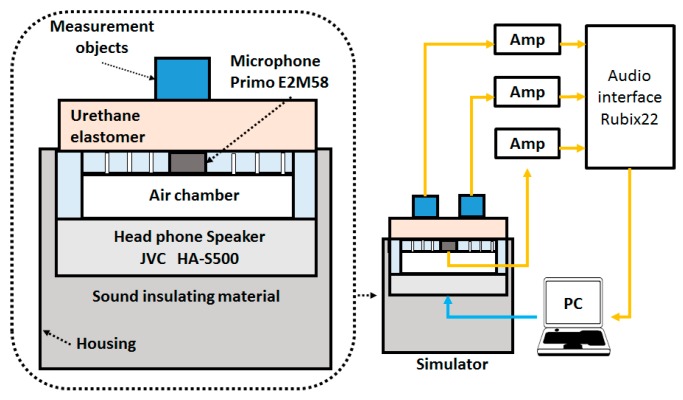
Respiratory sound simulator.

**Figure 5 sensors-20-00942-f005:**
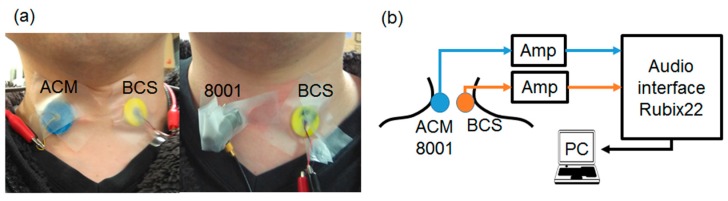
Actual tracheal sound observation system.(**a**) A sensor is attached to the test subject (ACM and BCS on the left, type 8001 and BCS on the right). (**b**) Diagram of the actual measurement system. (**c**) Mask with the built-in speaker. (**d**) Diagram of the mask sound source measurement system.

**Figure 6 sensors-20-00942-f006:**
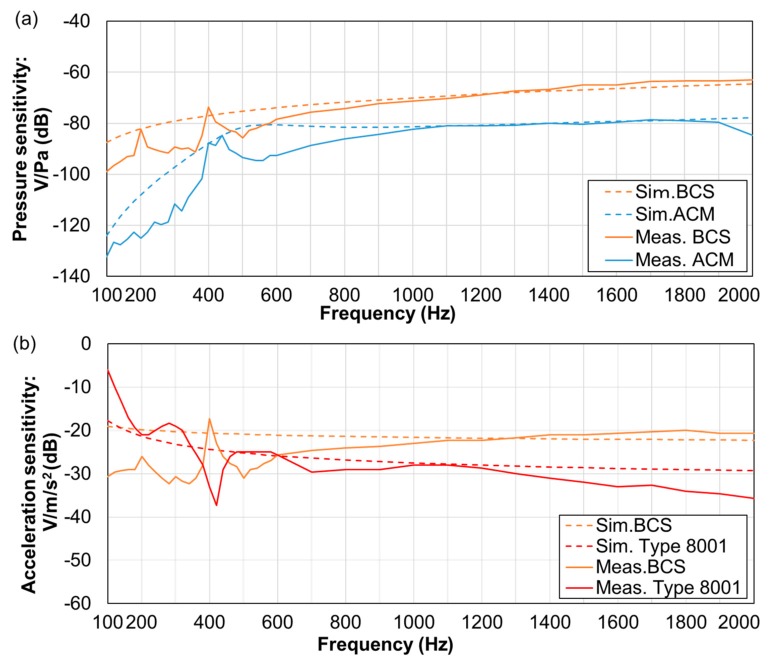
Frequency characteristics of sensors: (**a**) BCS and ACM voltage/pressure sensitivity. (**b**) BCS and type 8001 voltage/acceleration sensitivity.

**Figure 7 sensors-20-00942-f007:**
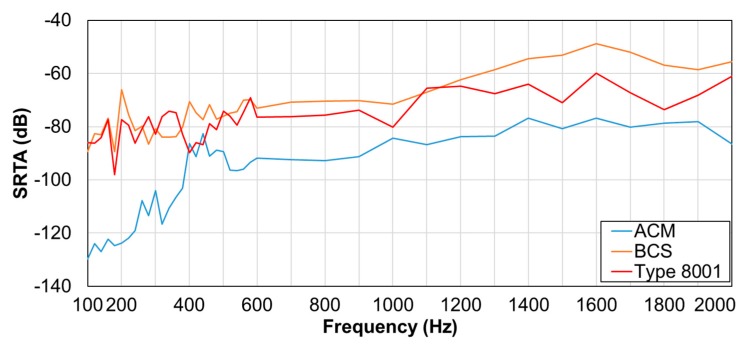
Tissue-borne sensitivity-to-air noise ratio (SRTA).

**Figure 8 sensors-20-00942-f008:**
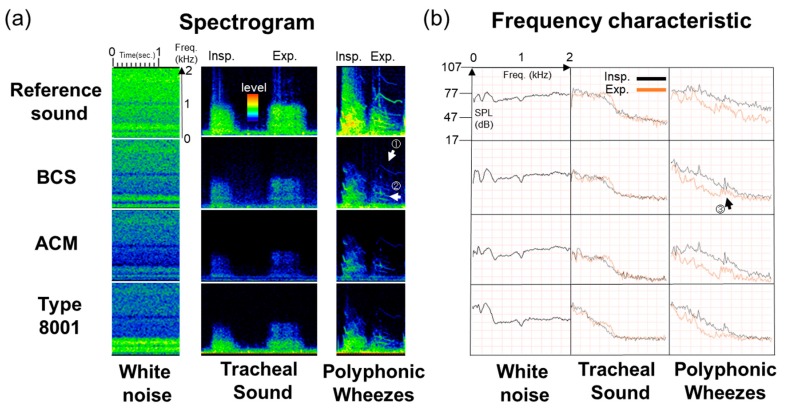
Observation experiment using a respiratory sound simulator. (**a**) Spectrograms and (**b**) Frequency responses of white noise, tracheal sound, and polyphonic wheezes.

**Figure 9 sensors-20-00942-f009:**
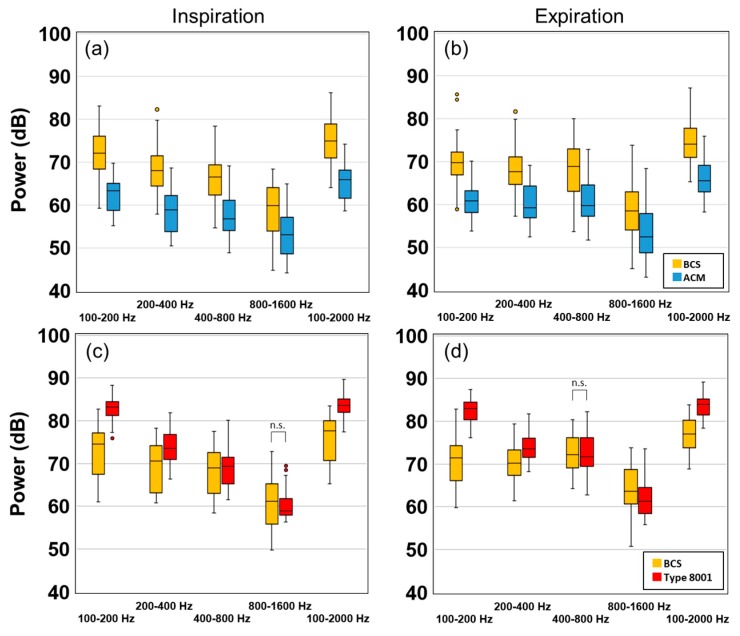
Experimental results of the subject's tracheal sound power values. (**a**) Inspiration of BCS and ACM, (**b**) Expiration of BCS and ACM, (**c**) Inspiration of BCS and type 8001, and (**d**) Expiration of BCS and type 8001. (n.s.: no significant difference)

**Figure 10 sensors-20-00942-f010:**
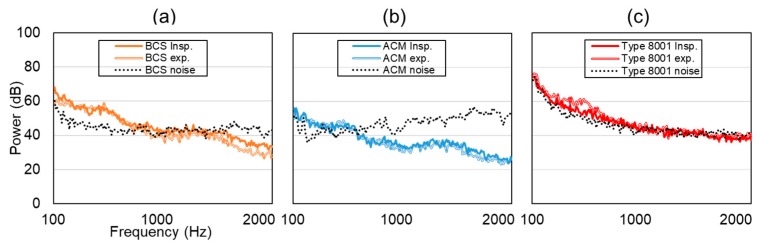
Experimental results of normal breathing tracheal sound in a noisy environment. (**a**) BCS, (**b**) ACM, (**c**) Type 8001, where the solid line represents the inspiration process, the double solid line is expiration, and the dotted line represents noise.

**Figure 11 sensors-20-00942-f011:**
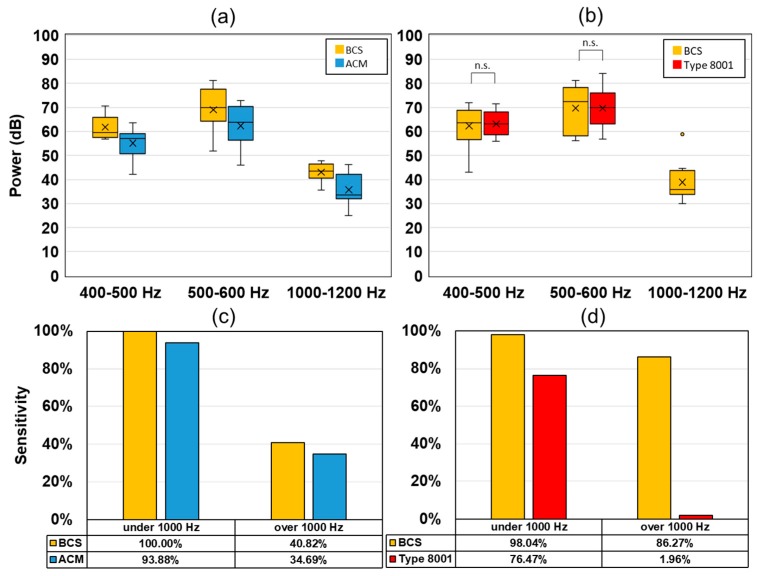
Experimental results of the wheezing sound from the oral transmission experiment. Wheezing detection power values for (**a**) BCS and ACM, (**b**) BCS and type 8001. Wheeze detection sensitivity (1000 Hz under/over), (**c**) BCS and ACM, and (**d**) BCS and type 8001.

**Figure 12 sensors-20-00942-f012:**
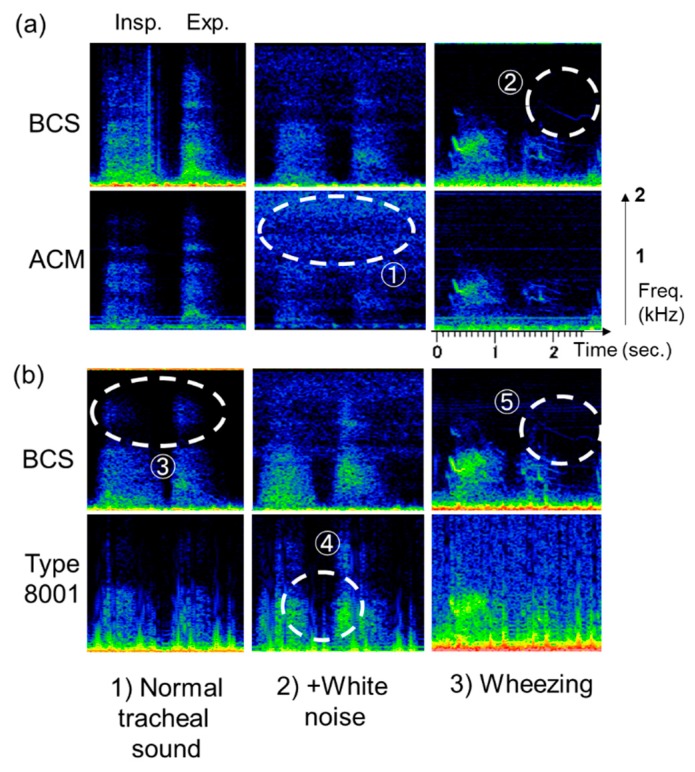
Respiratory sound detection in the sensors (representative cases). (**a**) BCS vs. ACM, (**b**) BCS vs. type 8001. Respiratory sounds observed included (1) normal breathing tracheal sound (left); (2) normal breathing tracheal sound in a noisy environment (center); and (3) wheezing sound from oral transmission (right).

**Table 1 sensors-20-00942-t001:** Specifications of the sensors.

	BCS	ACM	Type 8001
Diameter (mm)	20 ^*1^	20 ^*1^	15
Mass (g)	1.49	1.50	30.89
Sensitivity (dB)	−32 (V/Pa) ^*2^	−32 (V/Pa)	0 (V/m/s^2^) ^*3^
Sensor type	Microphone	Microphone	Accelerometer
Housing materials	ABS resin	ABS resin	-

*1 Diameter of the propagation layer, *2 Sensitivity of the remodeling microphone (EM258N-remodeling) contained in BCS, and *3 Sensitivity measurement includes the amplifier.

**Table 2 sensors-20-00942-t002:** Theoretical formula of sensors.

	BCS	ACM	Type 8001
Acceleration Response Model	$\frac{Z_{r}}{Z_{M}+Z_{r}}\;\left(1\right)$	−	$\frac{Z_{r}}{Z_{M}+Z_{r}}\;\left(2\right)$
Pressure Response Model	$\frac{Z_{r}}{Z_{M}+Z_{r}}A_{0}\left(\omega \right)\frac{m}{S}\;\left(3\right)$	$\frac{Z_{r}}{(Z_{m}+Z_{M})+Z_{r}}A_{0}\left(\omega \right)\;\left(4\right)$	−

**Table 3 sensors-20-00942-t003:** Parameters for test subjects.

Female/Male	3/12
Age (Mean ± SD)	24.6 ± 3.6
Height (cm) (Mean ± SD)	170.7 ± 10.0
Weight (kg) (Median (min–max))	66.0 (47.0–152.0)

**Table 4 sensors-20-00942-t004:** Power median (0.25–0.75%) value (dB) in each frequency segment.

**Tracheal Sound**					
		**100–200**	**200–400**	**400–800**	**800–1600**
BCS	Insp.	78.5 (75.0–80.0)	76.0 (71.0–79.0)	75.0 (68.3–78.8)	62.5 (62.0–63.0)
	Exp.	72.0 (71.3–73.5)	75.5 (74.3–77.0)	77.5 (74.0–78.8)	67.0 (63.3–69.0)
ACM	Insp.	70.5 (68.0–74.0)	64.0 (60.3–69.0)	63.0 (56.5–67.0)	48.0 (46.0–5 1.3)
	Exp.	67.5 (66.0–68.0)	64.5 (63.0–65.0)	66.0 (62.0–67.8)	57.0 (51.0–58.8)
Type 8001	Insp.	87.5 (84.8–89.8)	83.5 (82.3–86.8)	72.5 (69.0–74.8)	57.0 (56.3–58.0)
	Exp.	82.5 (79.5–83.0)	84.0 (81.3–84.8)	74.0 (70.8–75.8)	62.0 (60.0–65.0)
**Polyphonic Wheezes**					
		**100** **–** **200**	**200** **–** **400**	**400** **–** **800**	**800** **–** **1600**
BCS	Insp.	87.0 (83.3–89.8)	84.5 (80.5–88.8)	82.0 (79.5–87.0)	72.5 (68.5–73.0)
	Exp.	80.5 (76.0–84.8)	74.0 (70.3–77.8)	70.5 (65.0–73.8)	63.5 (63.0–64.0)
ACM	Insp.	71.5 (69.3–75.0)	76.5 (72.0–80.5)	75.0 (73.3–81.0)	67.0 (61.3–68.0)
	Exp.	66.0 (65.0–70.0)	60.0 (56.5–61.0)	56.5 (52.3–59.8)	52.0 (51.3–52.8)
Type 8001	Insp.	85.5 (84.0–91.0)	91.5 (87.5–93.0)	81.0 (80.0–86.5)	67.0 (65.3–68.0)
	Exp.	80.0 (76.0–84.0)	78.5 (73.5–81.3)	69.5 (58.0–74.0)	58.5 (58.0–59.0)

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
