# Peer review of "Validation of a Body-Conducted Sound Sensor for Respiratory Sound Monitoring and a Comparison with Several Sensors"

_sensors, 2020, doi:10.3390/s20030942_

Round 1

Reviewer 1 Report

The topic is of great interest in research as well as for practical applications. The authors compare different sensors in terms of their sensitivity. 

In the introduction a detailed overview of commercially available sensors including their characteristics and parameter is missing. In addition, more information regarding influencing factors would be necessary for a better understanding. Since we authors want to measure signals in humans, they should explain in more detail what the influence of the tissue is.

The authors focused their work only on the sensitivity of the used sensors. Sensors are characterized by much more parameter, e.g. measuring range, linearity, latency, drift etc.. An experimental evaluation and comparison of these parameter is completely missing. 

For the practical testing the authors only included 1 person. For a solid statistical evaluation of the measured data as well as for a real evaluation of the sensor more real data from humans would be required. This is especially important due to the great variability of human tissue. This is the main drawback of this paper. 

Reviewer 2 Report

See attached file for my comments

Reviewer 3 Report

The authors present a research on a candidate sensor for improved auscultation. The experimental study is well designed, including a simulator of noises to be recorded as well as  in vito measurements. The results are analysed and clearly presented. The conclusions are straight forward and well based on the results of the study.

The only major aspect that I find lacking is more information on the experiments. For example:
 - how many experiments were made?
 - how many different test subjects for the in vito experiments? Did their ages vary? Skin conductivity can vary with age.
 - were the environmental conditions varied at all (air humidity, skin humidity, temperature)
 - how were the results averaged?
 - more information on the standard deviation / error bars

The minor aspects regarding the study are:
 - what is the field of use for the presented sensors?
 - how long are typical measurements and what role does the length of measurement play in the results? For example - contact with skin can suffer from degradation in time. How sensitive are different sensors for this degradation? Note that this might be irrelevant for short measurements.
 - how sensitive is each sensor to sub-optimal placement?

Round 2

Reviewer 1 Report

Thank you for addressing my comments and including more experimental data of 15 volunteers. This really improved your article. 

However, I am still missing a good overview about existing systems including their parameter (not only the once you have used in your experiments). This should be considered in further publications.

In addition, I find it difficult to to focus only on the frequency characteristics of a sensor. Typical sensor parameter should always be part of an evaluation. 
